# JOM-4S Overhauser Magnetometer and Sensitivity Estimation

**DOI:** 10.3390/s21227698

**Published:** 2021-11-19

**Authors:** Xiaorong Gong, Shudong Chen, Shuang Zhang

**Affiliations:** College of Electronic Science and Engineering, Jilin University, Changchun 130012, China; gongxr21@mails.jlu.edu.cn (X.G.); chenshudong@jlu.edu.cn (S.C.)

**Keywords:** Overhauser magnetometer (OVM), sensitivity, proton magnetometer, dynamic nuclear polarization (DNP), scale quantum magnetometer

## Abstract

The Overhauser magnetometer is a scalar quantum magnetometer based on the dynamic nuclear polarization (DNP) effect in the Earth’s magnetic field. Sensitivity is a key technical specification reflecting the ability of instruments to sense small variations of the Earth’s magnetic field and is closely related to the signal-to-noise ratio (SNR) of the free induction decay (FID) signal. In this study, deuterated ^15^N TEMPONE radical is used in our sensor to obtain high DNP enhancement. The measured SNR of the FID signal is approximately 63/1, and the transverse relaxation time *T*_2_ is 2.68 s. The direct measurement method with a single instrument and the synchronous measurement method with two instruments are discussed for sensitivity estimation in time and frequency domains under different electromagnetic interference (EMI) environments and different time periods. For the first time, the correlation coefficient of the magnetic field measured by the two instruments is used to judge the degree of the influence of the environmental noise on the sensitivity estimation. The sensitivity evaluation in the field environment is successfully realized without electrical and magnetic shields. The direct measurement method is susceptible to EMI and cannot work in general electromagnetic environments, except it is sufficiently quiet. The synchronous measurement method has an excellent ability to remove most natural and artificial EMIs and can be used under noisy environments. Direct and synchronous experimental results show that the estimated sensitivity of the JOM-4S magnetometer is approximately 0.01 nT in time domain and approximately 0.01 nT/$\sqrt{\mathrm{Hz}}$ in frequency domain at a 3 s cycling time. This study provides a low-cost, simple, and effective sensitivity estimation method, which is especially suitable for developers and users to estimate the performance of the instrument.

## 1. Introduction

The proton magnetometer is a scalar quantum magnetometer that is based on the Larmor precession of hydrogen protons in the Earth’s magnetic field. In accordance with different polarization modes of hydrogen protons, proton magnetometers can be divided into two types: the first type is the classical proton precession magnetometer (PPM), which was first implemented by Packard, Varian, and Waters [1,2,3]. This type of magnetometer uses a bias magnetic field perpendicular to the Earth’s magnetic field to orient the hydrogen proton moment to the direction of the combined magnetic field. After quickly cutting off the polarizing current and the bias magnetic field disappears, the proton magnetic moment will rotate around the Earth’s magnetic field *B*_0_ with an angular velocity *ω* = *γ*_p_*B*_0_, and *B*_0_ can be obtained through an accurate calculation of *ω* [4,5]. The second type is the Overhauser magnetometer (OVM) that is based on the Overhauser effect researched by Overhauser, Abragam, and Solomon [6,7,8,9,10]. It uses the resonance of electrons to transfer to the resonance of the hydrogen nucleus, thereby enhancing its magnetic moment. The subsequent process is consistent with the PPM. Obviously, the OVM is similar to the PPM, except that the sensor design and the polarization strategies are different.

In the 1990’s, France [11,12,13], Canada [14], and Russia [15,16] developed the OVM. In recent years, China has conducted a series of studies on the development of the OVM [17,18,19,20,21,22]. Compared with the PPM, the OVM has the following advantages. The OVM uses electrons to orient the nuclear magnetic moment. Thus, it has an extremely high polarization efficiency. The power consumption is less than 2 W at a 3 s cycling time, which is especially suitable for portable measurement. The dynamic nuclear polarization (DNP) effect enhances the free induction decay (FID) signal so that the signal-to-noise ratio (SNR) can reach approximately 100, making the OVM more sensitive than the PPM. The lateral relaxation time *T*_2_ of the PPM is usually less than 1.0 s, whereas that of the OVM can usually reach 2.0 s or more due to the different working materials of the sensors. This condition greatly improves the SNR of the late signal and further improves the sensitivity. On the basis of the above advantages, the OVM is widely used in total magnetic field measurements, such as space magnetic detection [11,12,13,23], magnetic anomaly detection [24], mineral prospecting [24], gas pipeline detection [25], geophysical exploration [26], archaeological detection [27], and geomagnetic observatory [14].

Same as the classical PPM and the optically pumped magnetometer [28,29,30,31,32,33,34,35], the technical specifications of the OVM include resolution, sensitivity, (absolute) accuracy, measurement range, gradient tolerance, cycling rate, drift, temperature range, and power consumption. In these specifications, sensitivity is a key technical specification of magnetometers for practical applications. Sensitivity reflects the relative uncertainty of instrument readings, and the fluctuation of the magnetometer reading is similar to Gaussian white noise. This condition is why sensitivity is often named noise level and can be characterized by standard deviation (STD) or power spectral density (PSD) [11,12,13]. Theoretically, when the STD is greater than the change in the measured micromagnetic anomaly, the measured micro magnetic anomaly will be submerged in the reading noise and cannot be detected.

Sensitivity is a characteristic of the magnetometer itself and should be measured in an undisturbed environment [17,28,29,30,31]. However, the environment without electromagnetic interference (EMI) is difficult to obtain. Therefore, investigating the sensitivity estimation method in an unshielded environment is necessary. This method is convenient for magnetometer users and developers to evaluate the performance of the instrument [32,33,34,35].

The sensitivity of the OVM is approximately an order of magnitude higher than that of the PPM, which can usually reach 0.015 nT/$\sqrt{\mathrm{Hz}}$ at 1 Hz [24]. Determining the sensitivity of homemade and commercial instruments through simple comparisons is difficult due to its high sensitivity [19]. Choosing a relatively stable piece of data to estimate the sensitivity to some extent eliminates the influence of diurnal variation on the sensitivity estimation, but the residual low-frequency components and high-frequency interference components still affect the sensitivity estimation [18]. When instruments with different sensitivities are not synchronized, the enhanced noise caused by subtracting data with each other degrades the accuracy of sensitivity estimation [22]. The fourth-order difference method can well eliminate the interference of low-frequency components, such as diurnal variation, but still cannot eliminate the high-frequency component interference [36]. The zero magnetic space is usually used for sensitivity estimation. However, it is difficult to obtain for ordinary users and developers, and the device that generates the magnetic field brings noise, which deteriorates the sensitivity estimation [18,19]. In summary, the estimated sensitivity is often inaccurate because no systematic, scientific, and effective evaluation method is available. The correlation coefficient of measurement results is introduced to quantitatively evaluate the effect of EMI noise on the reading noise of the magnetometer and to solve these problems. The direct measurement method with a single instrument and the synchronous measurement method with two instruments are discussed for sensitivity estimation in time and frequency domains under different EMI environments and different time periods.

The JOM-4S OVM is a portable magnetometer developed by the College of Electronic Science and Engineering of Jilin University over a decade. This study first introduces the JOM-4S OVM and describes it in detail, including its sensor and electronics. The application conditions of the direct method and synchronous method are discussed by introducing the correlation coefficient. The sensitivity under noisy and quiet environments by using two estimation methods at time and frequency domains is estimated.

## 2. Physical Principles

The core of the OVM sensor is the working material containing electrons and hydrogen protons. The working principle of the OVM sensor can be summarized into three stages: (1) excited by the radio frequency (RF) electromagnetic field, the electrons in free radicals produce electron spin resonance (ESR); (2) electron resonance drives hydrogen protons in solvent to produce nuclear magnetic resonance (NMR); (3) Larmor precession of hydrogen nuclei in solvent.

### 2.1. ESR

When deuterated ^15^*N* nitrogen oxygen radical TEMPONE dwells in an ambient *B*_0_ magnetic field, which is in the oz direction, the Hamiltonian of the electron spin *S* and the ^15^*N* nuclear spin *K* can be described as [37]
(1)$$H=-\gamma_{S}B_{0}S_{z}-\gamma_{K}B_{0}K_{z}+AS\cdot K,$$
where *A* is a hyperfine coupling constant of the two-spin system. *γ*_S_ and *γ*_K_ are the gyromagnetic ratios of the electron and ^15^*N* nuclear, respectively. For the ^15^N TEMPONE nitroxide radical, *S* = 1/2, and *K* = 1/2. In the local field of ^15^*N*, the unpaired electrons in the free radical are hyperfine coupled with a ^15^*N* nucleus, and Zeeman splitting occurs in the ambient Earth’s field. In accordance with the Breit Rabi relation [38], in the presence of an external magnetic field, the energy levels of ^15^*N* radicals will split into four, and the energy level diagram is shown in Figure 1. The *σ* transition (blue arrow) and the *π* transition (red arrow) are excited by excitation in the same direction because the external magnetic field and excitation are perpendicular to the external magnetic field [37].

### 2.2. Solomon’s Equation

In low fields, an effective means to amplify the NMR signal is through DNP by paramagnetic impurities. The amplification is obtained through the magnetic coupling between the protons and the unpaired electrons of the radicals in the OVM sensor. The coupled system has four energy levels, as shown in Figure 2 [8].

The first sign refers to electrons, and the second to protons, where *W*_0_, *W*_2_, *W*_1_′, and *W*_1_ are the transient probabilities per unit time between the four states. *W*_1_′ denotes electronic transition, which realizes the transition of the electron from the eigenstate |−> to |+> (or |+> to |−>), and the process is completed by RF excitation. In the thermal equilibrium state, this transition will soon be saturated, but the two relaxation processes *W*_0_ and *W*_2_ ensure that the electron transitions from the eigenstate, |+> to |−> (or |−> to |+>). The electronic transition *W*_1_′ drives the proton *W*_1_′ to transition, realizing the transition of the proton from the eigenstate |−) to |+) (or |+) to |−)). The two relaxation processes *W*_0_ and *W*_2_ also ensure that the proton goes from the eigenstate|+) transition to |−) (or |−) to |+)), avoiding the saturation of the proton transition. The proton polarization is increased, and the direction is toward the geomagnetic field. The overall DNP enhancement of *E* is defined as [39]
(2)$$E=\frac{\langle I_{Z}\rangle }{I_{0}}=1-\xi \cdot s\cdot f\frac{\left|\gamma_{s}\right|}{\gamma_{n}},$$
where 〈*I_z_*〉 is the expectation value of the DNP, and *I*_0_ is its thermal equilibrium value. *γ_n_* and *γ_s_* are the gyromagnetic ratios of the nuclei and electron, respectively. *ξ*, *f*, and *s* are the coupling factor, leakage factor, and saturation factor, respectively. In the case of a pure scalar hyperfine dipolar coupling between the electronic spin *s* and the nuclear spin *I*, *ξ* = −1. The value of *f* determines the contribution of electrons to nuclear relaxation. In an ideal case, *f* = 1, the nuclei relaxation is completely controlled by radical electrons at this time. The saturation factor *s* is a measure of the saturation of the electronic transition. Ideally, the value of *s* is 1 when the electronic transition is fully saturated. Thus, in the ultralow field, the value that can be obtained by DNP is determined by *γ_s_*/*γ_k_*, and the theoretical DNF factor is 660 [39]. In low fields, such as the Earth’s magnetic field, several thousand times higher than its thermodynamic equilibrium can be obtained [14,15,16,17,18,19,20,21,22,23,24,25,26,27,36,37,38,39,40].

### 2.3. Lamor Precession

When the electron resonance transition is saturated, the resonant proton magnetic moment *M*_0_ will precess to the direction of the total magnetic field *B*(*B*_0_ + *B_p_*). Then RF polarization *B*_RF_ and DC polarization *B_p_* are removed, and the excited protons will precess around the Earth’s field *B*_0_, named the Larmor precession. During Larmor precession, the relation between the angular frequency of Larmor precession *ω*_0_ and the Earth’s field *B*_0_ is [4]
(3)$$\omega_{0}=\gamma_{p}B_{0},$$
where *γ**_p_* = 2.67515255 × 10^8^ T^−1^ s^−1^ is the gyromagnetic ratio of the hydrogen nucleus. As long as the frequency value of the Larmor signal is measured with a frequency meter, the Earth’s magnetic field can be accurately calculated.

## 3. Construction of JOM-4S OVM

### 3.1. Brief Introduction of JOM-4S OVM

As shown in Figure 3, the JOM-4S OVM consists of two parts, the instrument console and the sensor. The console includes four parts: analog board (Figure 4a), digital board (Figure 4b), key display board, and power board.

As shown in Figure 5, the Advanced RISC Machines (ARM) controls the Complex Programmable Logic Device (CPLD) to execute DC polarization, RF polarization, receiving, and tuning control. The sensor coil is shared by DC polarization and reception. The tuning capacitor and sensor coil form a LC parallel resonance to amplify the FID signal, and then the signal is sent to the last stage amplifier. The compared signal is sent to the CPLD to be counted. The count value is transmitted to the ARM to calculate the magnetic field. The rectified signal is sent to the A/D built-in ARM for signal quality evaluation.

### 3.2. Sensor Design

The heart of the OVM is the sensor, which is mainly composed of two parts: an RF resonant cavity and a low-frequency coil. The coaxial resonant cavity is used to obtain a highly uniform transverse circular polarization RF magnetic field to resonate the unpaired electron in the cavity. The low-frequency coil is sleeved outside the resonant cavity to generate a bias field and receive the FID signal. The components of the OVM sensor are shown in Figure 6.

Literature has shown that ^15^N-D-4-Oxo-2,2,6,6-tetramethylpiperidine-1-oxyl (^15^N-D-oxo-TEMPO) has the highest enhancement factor compared with radicals such as ^14^N-H-oxo-TEMPO and ^14^N-H-TEMPOL [40]. So ^15^N-D-oxo-TEMPO dissolved in solvent Diethylene glycol dimethyl ether (DME) is used in the JOM-4S Overhauser magnetometer for DNP enhancement. The molecular structure formula of ^15^N-D-oxo-TEMPO is shown in Figure 7.

The coaxial resonator is composed of a quartz bottle filled with a free radical solution and metal strips attached to the inside and outside of the container. Either the inner or outer conductor has gaps to prevent an eddy current. The length and the volume of the container are 112 mm and 175 mL, respectively. A simulation model for the coaxial resonant cavity is established by using Ansoft HFSS to study the magnetic field distribution of the resonant cavity under the RF field, as shown in Figure 8a. The magnetic field distribution is shown in Figure 8b. The resonant cavity generates a uniform circularly polarized magnetic field. The *S*11 parameter of the resonant cavity is measured with an Agilent 8712ES network analyzer (Santa Clara, CA, USA). As shown in Figure 8c, after carefully adjusting the matching network, the *S*11 parameter of the resonator is approximately −45 dB measured at 58.8 MHz; its −10 dB bandwidth covers approximately 58.823 MHz to 58.777 MHz, and the *Q* value is 1278.

The low-frequency coil is used for DC polarization and FID signal reception. The low-frequency coil adopts a differential double-coil structure composed of two inverted coils in series, and the distribution of the magnetic field in the coil after DC polarization is shown in Figure 9a. When EMI occurs in the external environment, the induced voltages generated by the two coils are the same in magnitude and opposite in phase due to the differential structure. These voltages can effectively remove external common mode interference.

The design of the sensor geometry is mainly considered to be as small as possible for obtaining a high gradient tolerance; excessive coil turns will lead to extremely large inductance, which will reduce coil bandwidth and portability; the decrease in coil turns is accompanied by the decrease in signal; extremely thick wire diameter indicates extremely large volume and loss of portability; extremely small wire diameter leads to extremely large resistance, thereby increasing the thermal noise of the sensor. Considering the geometric size, gradient tolerance, coil bandwidth, signal amplitude, and sensor noise, the resultant number of turns is 700; the wire gauge is 0.53 mm; the resistance is 22 Ω; the inductance is 35 mH; and the self resonant frequency is approximately 47 kHz after shielding. As shown in Figure 9b, the maximum value of the DC polarization field is designed to be 50 μT.

### 3.3. Analog Circuit Designs

The main task of the analog circuit is signal conditioning to obtain a volt-level FID signal with a high SNR. As shown in Figure 10, the analog circuit is divided into seven functions: DC polarization, RF polarization, preamplifier, buffer amplifier, post amplifier, rectifier, and comparator. The square wave outputted by the comparator is inputted to the CPLD for counting, and the envelope signal obtained by the rectifier is used for signal quality evaluation.

A tuning capacitor connected in parallel with the sensor coil is needed to cover the measurement range of 20–120 uT and to achieve a resonance circuit, as shown in Figure 11. *r* is the coil resistance, *L* is the coil inductance, *C* is the tuning capacitance, and *R_d_* is the matching resistance to limit the *Q* value for avoiding self-excited oscillation. LC-parallel resonance is an important method for improving the SNR of the FID signal. The *Q* values with and without *R_d_* are written in Equations (4) and (5).



(4)
$$Q^{\prime }=\frac{R_{d}}{Q^{2}r+R_{d}}Q$$


(5)
$$Q=\frac{L\omega }{r}$$



The calculated *Q* value of the sensor coil without *R_d_* in parallel is 22. After connecting *R_d_* in parallel, the *Q*′ value is reduced to 20.

The preamplifier is based on a low-noise amplifier with a 1.4 nV/$\sqrt{\mathrm{Hz}}$ voltage noise and a 0.1 pA/$\sqrt{\mathrm{Hz}}$ current noise at 1 kHz. High accuracy and temperature stability resistors are used throughout the amplifier chain to reduce the addition of Johnson noise. The last stage-amplified signal is sent to the comparator and rectifier and is then sent to the ARM for counting and signal quality evaluation. The test scheme of the indoor comparator circuit and rectifier circuit is shown in Figure 12. The 500 mV, 2.3 kHz frequency sinusoidal signal generated by the AFG-2225 Arbitrary Function Generator (Suzhou, China) is divided into 23 uV by 1.2 MΩ and 56 Ω resistors. The signals are measured with a DSOX3504T Digital Storage Oscilloscope (Santa Clara, CA, USA). As shown in Figure 13a, the TEST 1 output signal (yellow) with 3.5 V peak-to-peak values is shaped and rectified into a square signal (blue) and an envelope signal (green). Considering that the electromagnetic induction signal is proportional to the frequency and *Q* value of the sensor coil, a second-order filter circuit is used to filter out high-frequency noise and flat the transmission characteristics of the sensor and amplifier chain.

The FID signal is measured by a digital oscilloscope Rohde & Schwarz model RTH1002 (Munich, Germany) in the field environment shown in Figure 13b. The initial amplitude is 1.9 V, and the attenuation constant is 2.68 s. The STD of the measured noise is 30 mV, and the SNR is 1900/30 = 63, which is twice larger than that of the PPM.

### 3.4. Digital Circuit Designs

As shown in Figure 14, the low-power microcontroller STM32 is used as MCU in the digital part to realize measurement control, data transmission, magnetic field calculation, storage, and other functions by controlling the CPLD, LCD, key, Flash, RAM, GPS, RS232, and other peripheral devices. The FLASH storage space is 256 M-bit, which can store up to 0.5 M magnetic field values. The magnetometer can transmit measurement data to PC via RS232. RAM stores various initial setting states, such as the measurement mode. Key uses a 4 × 5 = 20 keyboard, LCD ZX1926M1A (Beijing, China) uses a 192 × 64 low-temperature liquid crystal with heating function, and the GPS error is 2.5 m. The CPLD mainly completes the measurement sequence control and counting.

JOM-4S is a slow-reading OVM. Its cycle time can be set from 3 s to 3600 s. As shown in Figure 15, the polarization time and reception time are set to 2 and 1 s for all cycle times, respectively. DC and RF polarizations are performed at the same time.

As shown in Figure 16, *T* is the preset gate time, which is controlled by signal quality. *T*_R_ is the actual measurement time and starts at the first rising edge in *T* and ends at the last rising edge after *T*. *N* is the number of cycles of the measured square wave signal, and *n* is the number of cycles of the reference signal generated by a 4 MHz temperature-compensated crystal oscillator (TCXO) with 2 ppm stability. *T*_R_ can be expressed by the following formula
(6)$$T_{R}=N/f_{s}=n/f_{0},$$
where *f_s_* is the frequency of FID signal. *f*_0_ is the frequency of the reference signal. Thus, the frequency of the measured signal can be calculated by
(7)$$f_{s}=Nf_{0}/n.$$

The ±1 count error of *n* can be ignored because *n* much-greater-than *N*.

## 4. Sensitivity Estimation Methods

As one of the important specifications of the OVM, sensitivity indicates the degree to which the measured value of the magnetometer deviates from the true value and can be calculated in terms of the STD of the reading noise of the instrument. In the environment with EMI, the measured total magnetic field *B*(*t*) can be expressed as the addition of several components.
(8)$$B(t)=B_{0}+B_{ln}(t)+B_{hn}(t)+B_{n}(t),$$
where *B*_0_ is the DC component of the Earth’s magnetic field, *B_ln_* is the low-frequency component consists of diurnal variation and drift, *B_hn_* is the high-frequency component consisting of environmental EMI, and *B_n_* is the reading noise of the OVM.

Theoretically, as long as *B_ln_*(*t*) = 0 and *B_hn_*(*t*) = 0 are satisfied, the sensitivity evaluation can be obtained by calculating the STD. However, this condition can only be satisfied in a zero magnetic environment. The following discussion focuses on the measurements in the field-unshielded environment where *B_ln_*(*t*) = 0 and *B_hn_*(*t*) = 0 are not satisfied. The direct measurement method and synchronization method are discussed to accurately evaluate the sensitivity in the time domain and spectral domain, respectively. Correlation analysis is used to evaluate the effect of the electromagnetic environment on sensitivity evaluation.

### 4.1. Fourth-Order Difference

When a single instrument is used to measure the Earth’s field in a field environment, low-frequency components, such as diurnal variation and other low-frequency interferences, are inevitably found in the measured total field. Low-frequency components can be filtered out to prevent the effect on sensitivity estimation because the fourth-order difference function has high-pass filtering characteristics. The fourth-order difference function can be expressed as
(9)$$T_{i}=B_{i-2}-4B_{i-1}+6B_{i}-4B_{i+1}+B_{i+2},$$
where *T_i_* is the fourth-order difference of the measured magnetic field *B_i_*.

After using the fourth-order difference method, we have *B_ln_*(*t*) ≈ 0. If high-frequency noise *B_hn_*(*t*) ≈ 0 can be satisfied, then sensitivity can be estimated accurately. However, *B_hn_*(*t*) existing in *B*(*t*) cannot be eliminated because its characteristic is similar to reading noise $B_{n}(t)$. Whether the environment meets the requirement *B_hn_*(*t*) ≈ 0 will be discussed in Section 4.2.

### 4.2. Correlation Analysis

After the fourth-order difference of the magnetic field, only high-frequency EMI noise *B_hn_* and the reading noise of the instrument *B_n_* are included in *B*(*t*). Theoretically, the high-frequency components of EMI noise *B_hn_* are the same in the magnetic field measured by the two synchronous instruments. We have
(10)$$T_{1}=T_{hn}+T_{n1},\;T_{2}=T_{hn}+T_{n2},$$
where 1 and 2 are the serial numbers of two instruments. *T*_1_ and *T*_2_ are the fourth-order differences of the magnetic field measured by two instruments. *T_hn_* is the fourth-order difference of the high-frequency components of EMI noise, and *T_n_*_1_ and *T_n_*_2_ are the reading noises of the two instruments. *T_hn_*, *T_n_*_1_, and *T_n_*_2_ are independent of each other, and the statistical characteristics of *T_n_*_1_ and *T_n_*_2_ are the same. Therefore, the correlation coefficients *ρ* of *T*_1_ and *T*_2_ can be expressed as
(11)$$\rho =\frac{Cov\left(T1,T2\right)}{\sqrt{D\left(T1\right)D\left(T2\right)}}=\frac{D\left(T_{hn}\right)}{D\left(T_{hn}\right)+D\left(T_{n1}\right)},$$
where *Cov* (*T*_1_, *T*_2_) is the covariance of *T*_1_ and *T*_2_, and *D*(*T*_1_), *D*(*T*_2_), *D*(*T_hn_*), and *D*(*T_n_*_1_) are the variances of *T*_1_, *T*_2_, *T_hn_*, and *T_n_*_1_, respectively. As shown in Equation (11), the correlation coefficients of *T*_1_ and *T*_2_ depend on the variance of the EMI noise *T_hn_* and the reading noise *T_n_*_1_ of the instrument. When the EMI noise *T_hn_* is greater than the *T_n_*_1_ of the instrument, the correlation coefficient is close to 1. When the EMI noise *T_hn_* is less than the reading noise *T_n_* of the instrument, the correlation coefficient is close to 0. Therefore, the correlation coefficient can be used as an indicator for judging whether the influence of the environmental noise level on the sensitivity estimation is negligible.

### 4.3. Direct Measurement Method in Time Domain

The direct measurement method estimates the instrument sensitivity by measuring the magnetic field through a single instrument. This method is usually used for testing and estimating the sensitivity value in the zero magnetic space [17]. However, the zero magnetic field space is difficult to obtain, and the field test is susceptible to the influence of the diurnal variation of the Earth’s field and EMI. Only a piece of approximately constant data can be usually selected for estimation so that the diurnal variation and external EMI can be approximately ignored. We use Equation (9) to perform the fourth-order difference on the measured data to eliminate low-frequency interference. In accordance with Equations (10) and (11), the correlation of the magnetic fields measured by two instruments is used to evaluate the influence of the high-frequency components of EMI noise *B_hn_*. When the reading noise of the instrument is dominant, coefficients *ρ* much-smaller-than 1, we have *B_hn_*(*t*) much-smaller-than *B_n_*(*t*) (or *B_hn_*(*t*) ≈ 0). In the time domain, STD is used to indicate sensitivity, as shown in Equation (12).
(12)$$\sigma =\frac{1}{\sqrt{70}}\sqrt{\frac{1}{n-1}\sum_{i=1}^{n}{\left(T_{i}-\bar{T}\right)}^{2}},$$
where $\bar{T}$ is the mean value of *T_i_*, and *T_i_* is the fourth-order difference of *B_i_*.

### 4.4. Synchronization Method in Time Domain

The synchronous measurement method refers to the simultaneous measurement of two instruments to obtain the sensitivity of one of the instruments. The synchronization method can remove the low-frequency components $B_{ln}(t)$ of the diurnal changes and effectively remove the high-frequency EMI $B_{hn}(t)$ during the subtraction process. Thus, we have *B_ln_*(*t*) ≈ 0 and *B_hn_*(*t*) ≈ 0. When the sensitivity of two sets is the same, the sensitivity of a single set can be expressed as Equation (13).
(13)$$\sigma =\frac{1}{\sqrt{2}}\frac{1}{\sqrt{70}}\sqrt{\frac{1}{n-1}\sum_{i=1}^{n}{\left(T_{1i}-T_{2i}+{\bar{T}}_{2i}-{\bar{T}}_{1i}\right)}^{2}},$$
where *T*_1*i*_ and *T*_2*i*_ represent the fourth-order difference of magnetic field values measured by the two instruments in the synchronous mode, and ${\bar{T}}_{1i}$ and ${\bar{T}}_{2i}$ are the mean values of *T*_1*i*_ and *T*_2*i*_, respectively.

### 4.5. PSD Estimation in Frequency Domain

In Section 4.3 and Section 4.4, the time domain sensitivity of the system is estimated by the STD of reading noise. In the frequency domain, the PSD of the magnetic field can be used to estimate the noise level of the instrument [11,17]. If the environment under test is sufficiently quiet, the fluctuation of the environmental magnetic field is lower than the noise of the magnetometer under testing. The spectral sensitivity can be calculated in accordance with Equation (14).
(14)$$PSD=\left|FFT(B(t)-\bar{B}(t))\right|,$$
where *B*(*t*) is the readings of the single magnetometer.

If the environment under test is insufficiently quiet, difference readings of two synchronous instruments are used to eliminate the interference of electromagnetic environment. Spectral sensitivity can be calculated in accordance with Equation (15).
(15)$$PSD=\frac{1}{\sqrt{2}}\left|FFT(B_{1}(t)-B_{2}(t)-{\bar{B}}_{1}(t)+{\bar{B}}_{2}(t))\right|,$$
where *B*_1_(*t*) and *B*_2_(*t*) are the readings of the two synchronized magnetometers, respectively. Either in Equation (14) or Equation (15), the DC components of the measured magnetic field are all removed to reduce their influence on high-frequency components. In Equation (15), the Hanning-windowed Fourier transform is used to prevent spectrum energy leakage.

## 5. Experimental Results and Discussions

The experiments were conducted in a quiet environment and noisy environment and lasted for approximately 24 h to observe the diurnal variations of the Earth’s magnetic field. The campus of Jilin University was chosen for the noisy environment experiment on 5 May 2021, and the rural area of the Jiutai District of Changchun City was chosen for the quiet environment experiment on 7 June 2021. The sensitivity estimation methods in the time and frequency domains were applied to evaluate the sensitivity of the JOM-4S magnetometer.

### 5.1. Sensitivity Estimation in Time Domain under a Quiet Environment

Two synchronous JOM-4S magnetometers were used to observe geomagnetic diurnal variation in the rural area of Jiutai City at a cycle rate of 3 s. The survey area was at the foot of Miaoxiangshan Mountain, far from the urban area, with a small EMI. Two instruments observed the value of the magnetic field for 24 h. The fourth-order difference of the magnetic field and the difference value of the fourth-order difference are shown in Figure 17.

As shown in Figure 17a, the diurnal variation of the magnetic field reached the maximum at 3–5 p.m. and the minimum at 9–11 a.m. The whole diurnal variation fluctuated by approximately 50 nT. As shown in Figure 17b, the fourth-order difference of the two magnetic field values completely eliminates the diurnal variation. From 10:00 p.m. to 4:00 a.m., the peak value of the fourth-order difference of the magnetic field was less than 1.0 nT. In the rest of the time, the peak-to-peak value of the fourth-order difference of the magnetic field reached 5–6 nT. As shown in Figure 17c, the correlation coefficient of the fourth-order difference of S001 and S002 was less than 0.2 between 10:00 p.m. and 4:00 a.m., but near 1 at other times. This condition indicates that the reading noise *B_n_*(*t*) of the instrument dominated between 10:00 p.m. and 4:00 a.m. The sensitivity of the instrument was estimated to be 0.0090 and 0.0092 nT with a direct measuring method between this period. In the rest of the time, the peak-to-peak value of the fourth-order difference of the magnetic field reached 5–6 nT, and the sensitivity estimated by the direct measuring method was 0.0417 and 0.0418 nT, which was greater than in the night. As shown in Figure 17d, two fourth-order differences were made to estimate the target sensitivity more accurately. The 24-h data and the 7-h data from 10:00 p.m. to 4:00 a.m. were applied with synchronization method to estimate the sensitivity as 0.0090 and 0.0086 nT, respectively.

The sensitivity of the magnetometer can be accurately estimated with the direct measuring method at night under a quiet environment as 0.009 nT or with the synchronization method throughout 24 h as 0.009 nT.

### 5.2. Sensitivity Estimation in Time Domain under a Noisy Environment

Long-time measurement (24 h) was conducted on the campus of Jilin University on 5 May 2021 at the same cycle rate of 3 s to analyze the influence of noise on the sensitivity evaluation of the magnetometer system. The campus environment is extremely complex, and the measurement results are affected by interference, cable interference, and automobile interference. Two instruments observed the value of the magnetic field for 24 h. The fourth-order difference of the magnetic field and the difference value of the fourth-order difference are shown in Figure 18.

As shown in Figure 18, the magnetic field fluctuated up to 100 nT in a short time during the day under high noise conditions. From 10:00 p.m. to 4:00 a.m., the fluctuation was extremely low. Therefore, the estimation of system sensitivity can only be conducted between 10:00 p.m. and 4:00 a.m. As shown in Figure 18b, the STDs of the fourth-order difference of the two magnetic field values reached 3.8684, 3.8677 nT in the daytime and 0.4239 nT and 0.4217 nT from 10:00 p.m. to 4:00 a.m. Specifically, even at night, the fourth-order difference of the magnetic field value in the high-noise environment still reflected the environmental noise and cannot be used to estimate the sensitivity. As shown in Figure 18c, the correlation coefficient of the fourth-order difference of S001 and S002 was extremely near to 1 throughout the whole day. In other words, the EMI noise was greater than the reading noise of the instrument. Thus, the sensitivity of the instrument cannot be estimated by using the direct measurement method. As shown in Figure 18d, the synchronization method was chosen to estimate the sensitivity in accordance with Equation (13). Taking the data of 7 h from 10:00 p.m. to 4:00 a.m., the sensitivity of the system is estimated to be 0.0084 nT.

### 5.3. Sensitivity Estimation in Frequency Domain

In accordance with the time domain analysis in Section 5.2, the EMI intensity was the lowest between 10:00 p.m. and 4:00 a.m. in a day. Thus, the magnetic field data in this time period are selected to estimate sensitivity. The direct spectrum estimation method of sensitivity was obtained by substituting the S001 or S002 time domain data in Equation (14) to calculate the PSD. The synchronous spectrum estimation method of sensitivity substitutes the time domain data difference of two instruments in Equation (15) to calculate the PSD.

As shown in Figure 19a, the PSDs of S001 and S002 in the low-frequency band were highly consistent. The form of PSD of S001–S002 was consistent with S001 and S002, but the amplitude was reduced by one order of magnitude due to the subtraction operation. These findings indicate that the diurnal variation and low-frequency interference are dominant in the low-frequency band and do not reflect the noise level of the instrument. In the high-frequency band between 0.3–0.5 Hz, the PSD was enlarged, as shown in Figure 19b. The PSDs of S001, S002, and S001–S002 are independent of each other and tend to be 0.01 nT/$\sqrt{\mathrm{Hz}}$. This value is consistent with the case when the correlation coefficient is 0.2 in the time domain. This condition implies that the noise level of the instrument is dominant in the high-frequency band. The sensitivity estimation can be obtained by either the synchronous method or the direct method under a quiet environment, and 0.01 nT/$\sqrt{\mathrm{Hz}}$ is the frequency domain sensitivity of JOM-4S magnetometer.

As shown in Figure 19c,d, the same sensitivity evaluation experiments were implemented in the noisy environment. Different from the results in the quiet environment, S001 and S002 coincide in all frequency bands, indicating that the measured data of the two instruments are highly correlated. This condition is consistent with the case when the correlation coefficient is 0.999 in the time domain. Therefore, the direct estimation method fails in noisy environment, and the sensitivity evaluation can only be conducted through the synchronization method because the EMI noise is dominant at all frequency bands. The synchronization method can effectively eliminate the low-frequency and high-frequency interferences in the environment and obtain the same evaluation results as in the quiet environment.

## 6. Conclusions

The JOM-4S OVM, developed by the College of Electronic Science and Engineering of Jilin University, is briefly introduced, including its sensor and circuit. JOM-4S is a portable slow-reading OVM magnetometer, and its cycle time can be set from 3 s to 3600 s. TEMPONE radical is used as an electron nuclear double-resonance working material in the proton-rich resonant cavity and significantly improves the DNP effect. The improved *Q* value of resonant cavity can reach 1278, resulting in the initial signal amplitude of 1.9 V. The SNR of the FID signal can reach to 63/1 when the 1 W RF output power is polarized. The transverse relaxation time *T*2 can reach 2.68 s, greatly improving the average SNR of the FID signal in one cycle. In this study, the low-noise sensor, low-noise signal conditioning circuit, and equal-precision counting method jointly ensure that the JOM-4S magnetometer has an extremely high sensitivity.

The feasibility of the accurate sensitivity evaluation of the magnetometer in the field environment is discussed comprehensively through theoretical analysis and experimental verification. In the field environment, the nature of various noises that affect sensitivity evaluation and the mechanism of how these noises affect the accuracy of sensitivity estimation are investigated. A synchronization method using two instruments and the direct method using a single instrument can achieve accurate sensitivity estimation under the condition that the influence of environmental noise on the reading noise of the instrument is negligible. As a quantitative evaluation standard of the electromagnetic environment, the correlation coefficient of the reading noise of the two instruments is proposed. The analysis results show that the direct method cannot accurately estimate the sensitivity when the correlation coefficient approaches 1 but can accurately estimate the sensitivity when the correlation coefficient approaches 0.

The sensitivity estimation experiments in the field environment show that the synchronization method can effectively eliminate most of the low-frequency and high-frequency EMIs in the environment. Thus, the synchronization method can achieve accurate sensitivity estimation as long as it is not in an extremely noisy environment. The experimental results show that only in an extremely quiet field environment can the correlation coefficient be close to 0, and the sensitivity of the magnetometer can be accurately estimated by using the direct method. All experimental results agree well with the theoretical predictions. Thus, the sensitivity evaluation of the magnetometer can be evaluated in the field environment. This method is simple, effective, and low cost.

Sensitivity estimations in time and frequency domains are conducted to estimate the sensitivity in different EMI environments. The results show that the time domain sensitivity and the frequency domain sensitivity of the JOM-4S OVM are approximately 0.01 nT and 0.01 nT/$\sqrt{\mathrm{Hz}}$ at a 3 s cycle rate, respectively. The sensitivity of the JOM-4S Overhauser magnetometer introduced in this article is comparable to that of the commercial magnetometer. Therefore, the JOM-4S OVM can measure the Earth’s magnetic field with sufficient sensitivity and can be used in various fields of weak magnetic measurement.

## Figures and Tables

**Figure 1 sensors-21-07698-f001:**
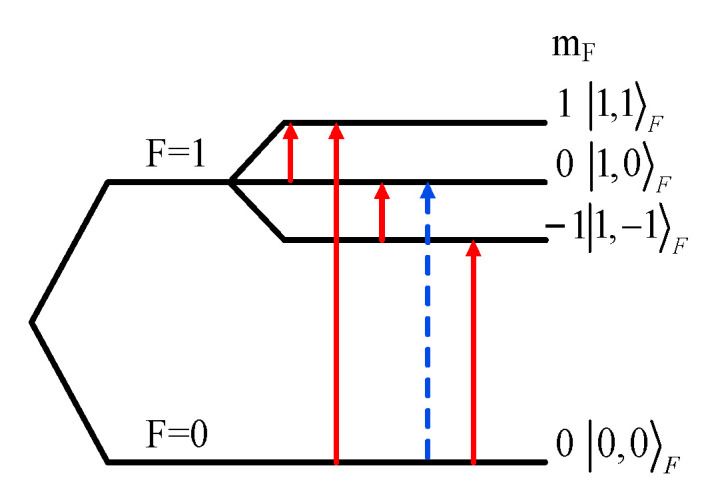
Energy levels of ^15^*N* nitroxide radical.

**Figure 2 sensors-21-07698-f002:**
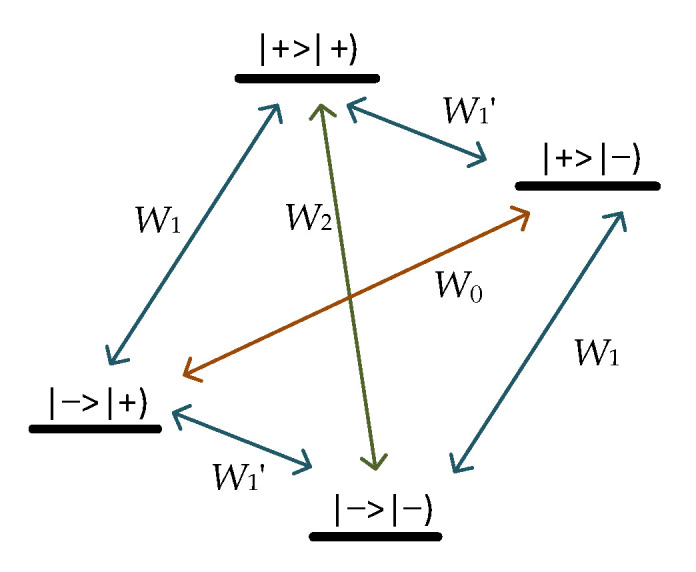
Energy diagram appropriate for coupled proton and electron, where |+>|+), |−>|−), |+>|−), and |−>|+) are the four energy levels, respectively.

**Figure 3 sensors-21-07698-f003:**
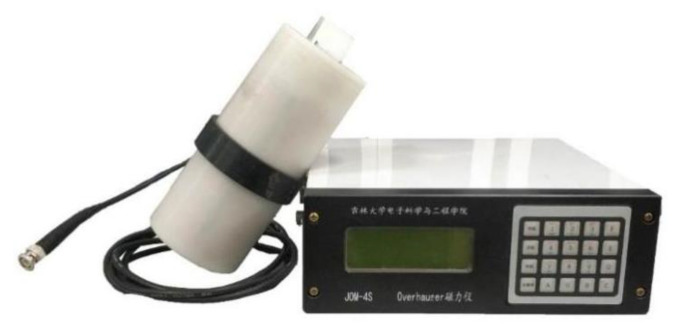
JOM-4S OVM.

**Figure 4 sensors-21-07698-f004:**
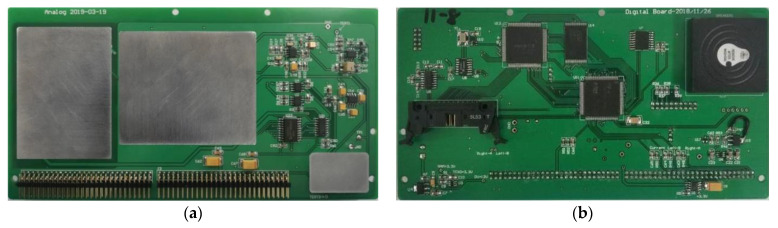
Analog board (**a**) and digital board (**b**) of JOM-4S OVM.

**Figure 5 sensors-21-07698-f005:**
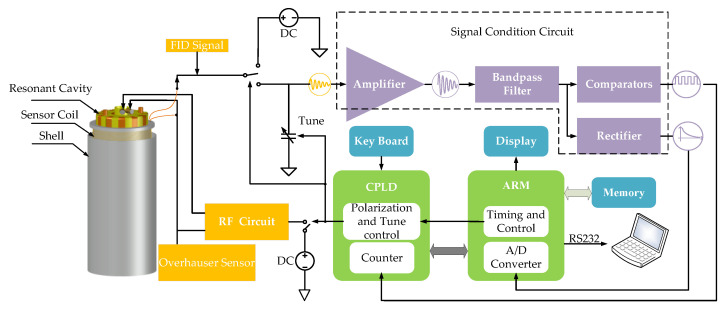
Block diagram of JOM-4S OVM.

**Figure 6 sensors-21-07698-f006:**
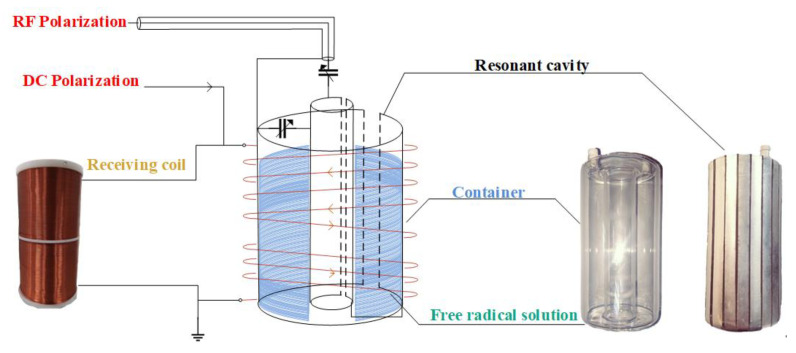
Components of the sensor.

**Figure 7 sensors-21-07698-f007:**
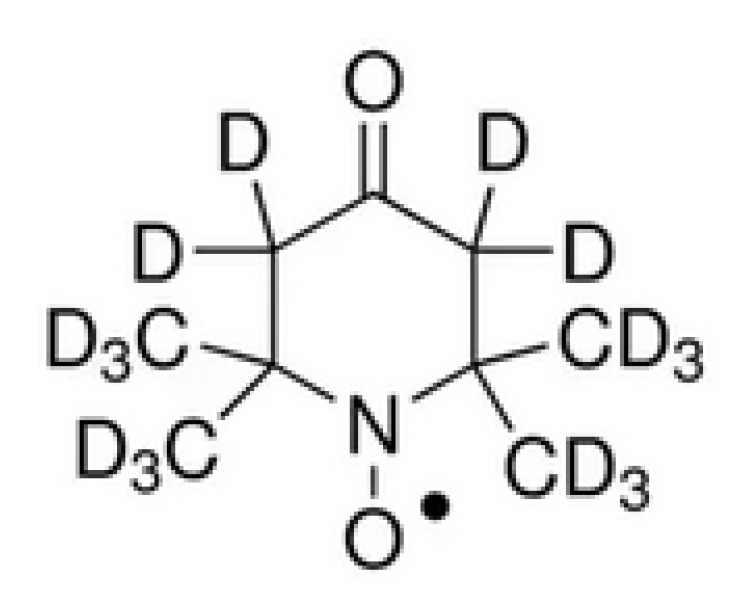
Molecular structure formula of ^15^N-D-oxo-TEMPO.

**Figure 8 sensors-21-07698-f008:**
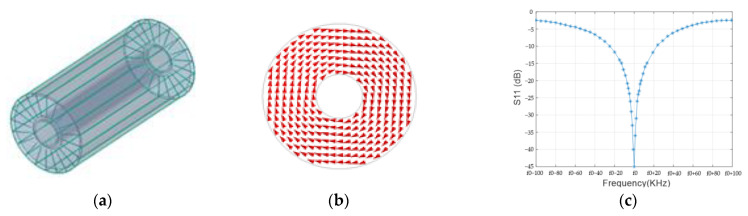
Coaxial cavity HFSS model (**a**), cross sectional magnetic field distribution (**b**) and *S*11 of resonant cavity (**c**).

**Figure 9 sensors-21-07698-f009:**
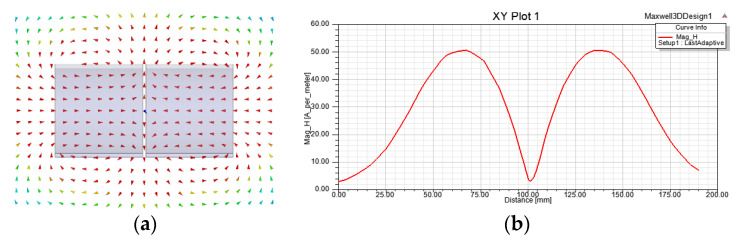
MAXWELL simulation of low-frequency coil (**a**) and axial magnetic field intensity distribution (**b**).

**Figure 10 sensors-21-07698-f010:**
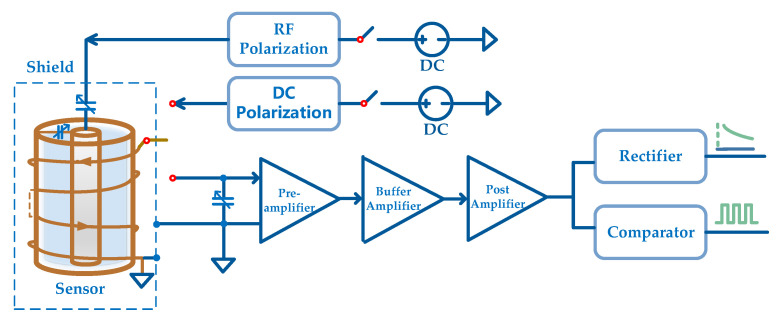
Block diagram of analog circuit.

**Figure 11 sensors-21-07698-f011:**
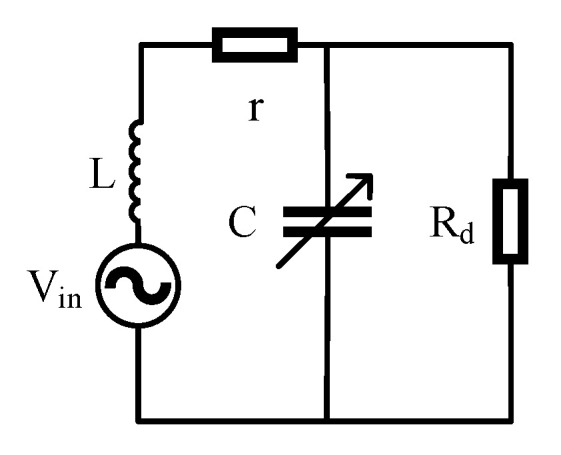
Equivalent circuit of sensor coil.

**Figure 12 sensors-21-07698-f012:**
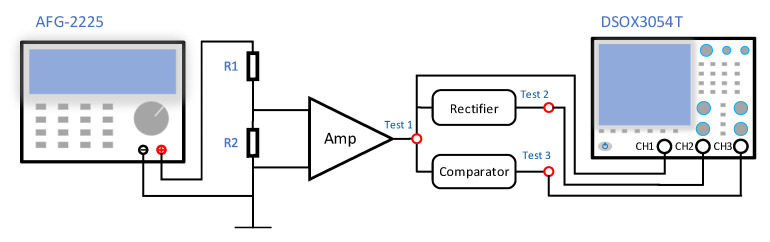
Schematic of envelope signal and shaping signal test.

**Figure 13 sensors-21-07698-f013:**
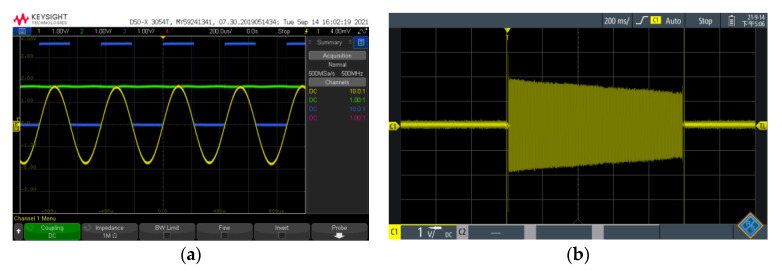
(**a**) Simulated FID signal, shaped signal, and envelope, (**b**) Measured FID signal.

**Figure 14 sensors-21-07698-f014:**
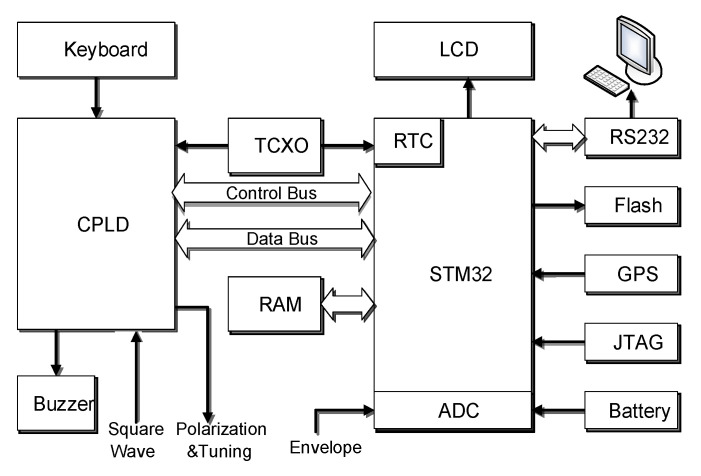
Digital circuit block diagram.

**Figure 15 sensors-21-07698-f015:**
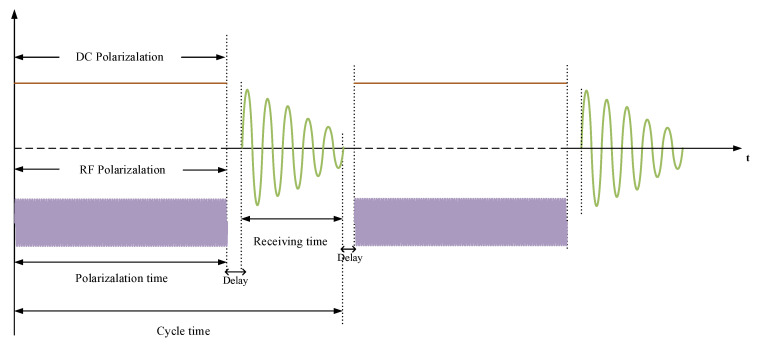
Sequence of polarization and reception controlled by ARM and CPLD.

**Figure 16 sensors-21-07698-f016:**
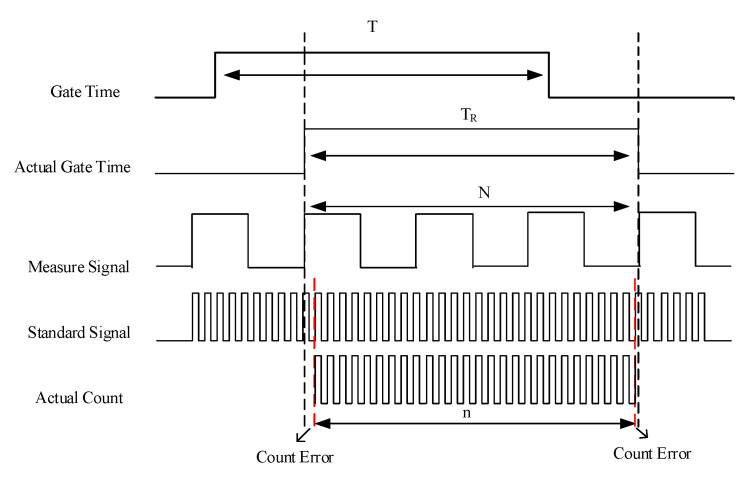
Schematic of frequency measurement.

**Figure 17 sensors-21-07698-f017:**
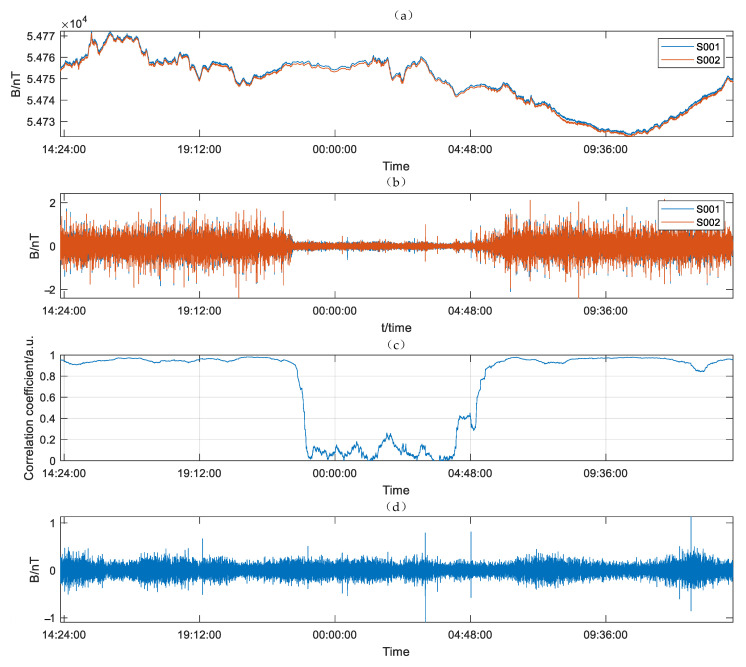
(**a**) 24 h magnetic field value, (**b**) fourth-order difference of the magnetic field, (**c**) correlation coefficient of S001 and S002, (**d**) difference value of the fourth-order difference.

**Figure 18 sensors-21-07698-f018:**
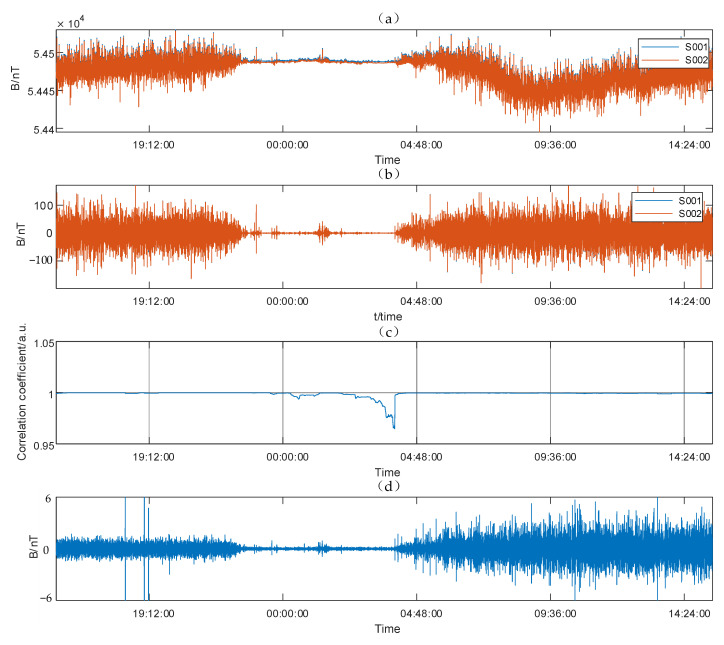
(**a**) 24 h magnetic field value, (**b**) fourth-order difference of the magnetic field, (**c**) correlation coefficient of fourth-order difference of S001 and S002, (**d**) difference value of the fourth-order difference.

**Figure 19 sensors-21-07698-f019:**
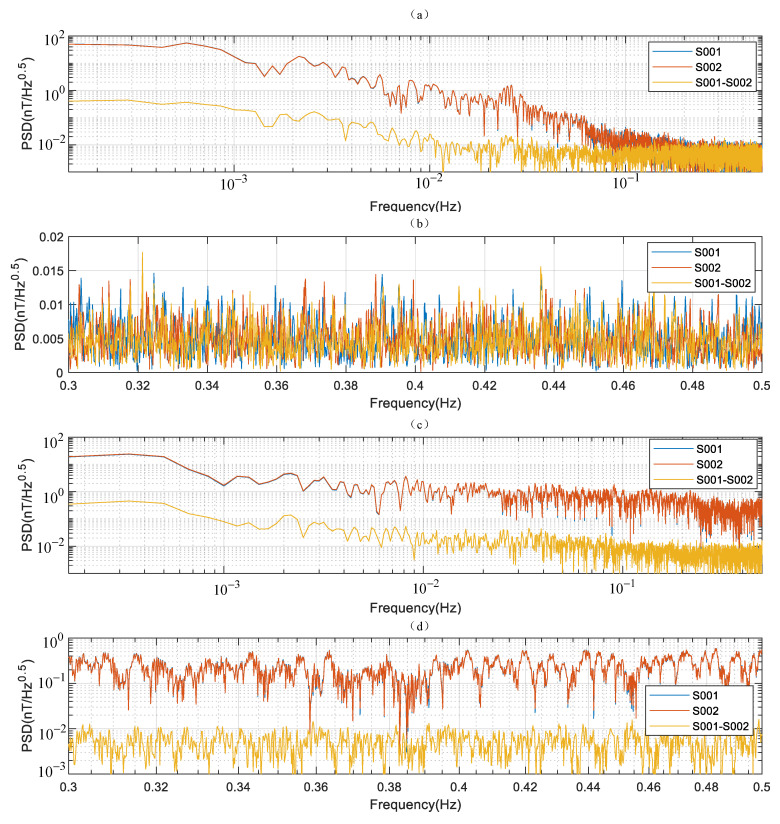
(**a**) PSDs under quiet environment, (**b**) high-frequency components of PSDs under quiet environment, (**c**) PSDs under noisy environment, (**d**) high-frequency components of PSDs under noisy environment.

## Data Availability

No new data were created or analyzed in this study. Data sharing is inapplicable to this article.

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
