# Peer review of "JOM-4S Overhauser Magnetometer and Sensitivity Estimation"

_sensors, 2021, doi:10.3390/s21227698_

Round 1
Reviewer 1 Report
The authors present the description and result of testing of the Overhauser magnetometer (the model JOM-4S ). The Overhauser magnetometer, with its unique set of features, represents a pillar of modern magnetometry of the Earth’s magnetic field. It also offers superior omnidirectional sensors; no dead zones; no heading errors; or warm-up time prior to surveys; wide temperature range of operation. The synchronous measurement method with two instruments was developed and investigated for sensitivity estimation in time and frequency domains. The results show that the estimated sensitivity of the developed magnetometer is approximately 0.01 nT in time domain and approximately 0.01 nT/√Hz in frequency domain at 3 s cycling time. The manuscript present new results which could be interesting for practical application.
Nevertheless the are some misprints and ambiguities which should be corrected.
In the line 106 it is written "...excited by radio frequency (RF) magnetic field...", but magnetic field can't change the energy of the system, so it must be written "...excited by radio frequency (RF) electromagnetic field...".
In the figure 2 the is no W1' which descibes in the text of the manuscript.
The statement in the line 142 "The proton magnetic moment is enhanced..." is wrong - the proton magnetic moment is constant and can't be change. The correction must be made.
In the equation (3) the index 0 must be added to magnetic induction B.
In the line 177 the abbreviations ARM and CPLD are not described.
In the figure 6 the misprint in the word "resonant" must be corrected.
In the lines 221-222 it is written, that "As shown in Figure 7(b), the maximum value of DC polarization field is designed to be 50 μT." But thereis no such date in figure 7(b). So either figure 7(b) or the text must be corrected.
In the figure 6 a part is named "post amplifier", but in the text (line 226) the expression "last stage amplifier" is used. The correction must be made either in the figure or in the text.
In the line 265 it is written "...low-power STM32 is used...", I would recommend to write "...low-power microcontroller STM32 is used...".
In the lines 270-271 it is written "LCD uses 192×64 low-temperature liquid crystal with heating function" - it is not clear. The type and manufactor of the LCD must be written.
In the equation (12) the number 70 must be explained - how it was calculated.
The figure 16(a) - the presented curve is very similar to the day temperature change. So the device temperature stability must be analysed.
The electronic scheme of device produce the magnetic field also - its contributions should discussed also.
Reviewer 2 Report
The authors must describe and characterize the TEMPONE material and explain why it gives superior results.
The authors must furnish results for other Overhauser sensors for comparison with their results. This information must be placed in the Conclusion section of the paper.
